# Chemical Composition, Antioxidant, Antibacterial, and Antibiofilm Activities of *Backhousia citriodora* Essential Oil

**DOI:** 10.3390/molecules27154895

**Published:** 2022-07-31

**Authors:** Ann Chie Lim, Shirley Gee Hoon Tang, Noraziah Mohamad Zin, Abdul Mutalib Maisarah, Indang Ariati Ariffin, Pin Jern Ker, Teuku Meurah Indra Mahlia

**Affiliations:** 1School of Graduate Studies, Management and Science University, University Drive, Off Persiaran Olahraga, Section 13, Shah Alam 40100, Malaysia; angeline.annqi@gmail.com (A.C.L.); maisarah_abdulmutalib@msu.edu.my (A.M.M.); 2International Medical School, Management and Science University, University Drive, Off Persiaran Olahraga, Section 13, Shah Alam 40100, Malaysia; indang@msu.edu.my; 3Center for Toxicology and Health Risk Studies (CORE), Faculty of Health Sciences, Universiti Kebangsaan Malaysia, Jalan Raja Muda Abdul Aziz, Kuala Lumpur 50300, Malaysia; 4Center of Diagnostics, Therapeutics & Investigations, Faculty of Health Sciences, Universiti Kebangsaan Malaysia, Jalan Raja Muda Abdul Aziz, Kuala Lumpur 50300, Malaysia; noraziah.zin@ukm.edu.my; 5Institute of Sustainable Energy, Department of Electrical and Electronics Engineering, Universiti Tenaga Nasional, Kajang 43000, Malaysia; pinjern@uniten.edu.my; 6School of Civil and Environmental Engineering, University of Technology Sydney, Sydney, NSW 2007, Australia; tmindra.mahlia@uts.edu.au

**Keywords:** *Backhousia citriodora*, essential oil, chemical composition, antioxidant, antibacterial, antibiofilm

## Abstract

The essential oil of *Backhousia citriodora*, commonly known as lemon myrtle oil, possesses various beneficial properties due to its richness in bioactive compounds. This study aimed to characterize the chemical profile of the essential oil isolated from leaves of *Backhousia citriodora* (BCEO) and its biological properties, including antioxidant, antibacterial, and antibiofilm activities. Using gas chromatography–mass spectrometry, 21 compounds were identified in BCEO, representing 98.50% of the total oil content. The isomers of citral, geranial (52.13%), and neral (37.65%) were detected as the main constituents. The evaluation of DPPH radical scavenging activity and ferric reducing antioxidant power showed that BCEO exhibited strong antioxidant activity at IC_50_ of 42.57 μg/mL and EC_50_ of 20.03 μg/mL, respectively. The antibacterial activity results showed that BCEO exhibited stronger antibacterial activity against Gram-positive bacteria (*Staphylococcus aureus* and *Staphylococcus epidermidis*) than against Gram-negative bacteria (*Escherichia coli* and *Klebsiella pneumoniae*). For the agar disk diffusion method, *S. epidermidis* was the most sensitive to BCEO with an inhibition zone diameter of 50.17 mm, followed by *S. aureus* (31.13 mm), *E. coli* (20.33 mm), and *K. pneumoniae* (12.67 mm). The results from the microdilution method showed that BCEO exhibited the highest activity against *S. epidermidis* and *S. aureus*, with the minimal inhibitory concentration (MIC) value of 6.25 μL/mL. BCEO acts as a potent antibiofilm agent with dual actions, inhibiting (85.10% to 96.44%) and eradicating (70.92% to 90.73%) of the biofilms formed by the four tested bacteria strains, compared with streptomycin (biofilm inhibition, 67.65% to 94.29% and biofilm eradication, 49.97% to 89.73%). This study highlights that BCEO can potentially be a natural antioxidant agent, antibacterial agent, and antibiofilm agent that could be applied in the pharmaceutical and food industries. To the best of the authors’ knowledge, this is the first report, on the antibiofilm activity of BCEO against four common nosocomial pathogens.

## 1. Introduction

Antibiotics, which are also called antibacterial or antimicrobial drugs, have significantly improved human health and life expectancy by preventing or treating various infectious diseases. However, antimicrobial resistance (AMR) is still a huge public concern, resulting in great loss to individual and social economies [1]. It has been estimated that the death rate due to AMR will rise to 10 million lives per year, with costs of USD 100 trillion, by 2050 [1,2]. In 2020, World Health Organization (WHO) released a report that identified a lack of innovation in the development of new antibiotics to combat the spread of drug-resistant bacteria in humans [3]. WHO declared AMR as one of the top 10 global public health threats facing humanity [4]. The misuse and abusive uses of antibiotics are the main drivers for the development of multidrug-resistant pathogens. Today, the rapid development of AMR presents a challenge in the treatment of infectious diseases. AMR reduces the number of therapeutic options, increases the hospitalization costs for patients, and increases morbimortality rates [5]. Thus, urgent multisectoral actions to tackle AMR are required to achieve the United Nations’ Sustainable Development Goals (SDGs). 

Another crucial public health issue is the ability of some bacteria strains to form complex multistructural biofilms [6]. Biofilms are tridimensional networks of bacterial cells that are entangled in a self-generated extracellular polymeric matrix composed of proteins, polysaccharides, lipids, and nuclei acids [7]. Biofilm architecture has hindered the penetration of antibiotics, reducing their capacity to reach internal layers and rendering them ineffective [8]. The resistance of bacterial biofilms to antibiotics leads to the persistence of infectious diseases, especially those diseases that are associated with the indwelling and implanting of medical devices [9]. Oxidative stress is characterized as a phenomenon caused by an imbalance between the production of reactive species and antioxidant defense activity; its enhanced state is strongly associated with various human chronic and infectious diseases [10]. Meanwhile, synthetic antioxidants, such as benzotriazole (BTA) and butylated hydroxyanisole (BHA) used in the food industry for food stabilization, have been found in animal and human studies to be responsible for carcinogenesis and toxicity [11,12,13,14]. Hence, the tendency to substitute these synthetic antioxidants with natural ones to control the signaling production of intracellular reactive oxygen species (ROS) and ROS-induced diseases has increased.

Recently, essential oils (EOs) have received increasing attention due to their promising bioactive compounds, which have various therapeutic properties and chemical diversity that are beneficial to health. EOs have been generally recognized as safe (GRAS) for human consumption by the United States Food and Drug Administration (FDA), when used for their intended purpose [15]. EOs are aromatic, volatile liquids that are extracted from plant materials. They are mainly composed of lipophilic and highly volatile secondary plant metabolites, including monoterpene and sesquiterpene hydrocarbons, oxygenated derivatives, derivatives of phenyl propane and phenols, and other volatile organic compounds [16,17,18]. EOs have been recognized as exhibiting antioxidant, antimicrobial, antiviral, antiparasitic, antifungal, anti-nociceptive, anti-inflammatory, and insecticidal activities [19,20,21,22,23,24,25,26]. Various studies have reported that the biological activities of EOs are attributable to major compounds; however, it has also been demonstrated that these compounds can interact with minor compounds, leading to synergistic or antagonistic effects and influencing the properties of EOs [27,28]. Consequently, the biological activities of EOs can differ according to variations in their chemical profile. Therefore, investigating the chemical composition and biological activities of EOs is necessary to confirm their use in various industries, including the pharmaceutical, nutraceutical, cosmetics, and agriculture industries.

*Backhousia citriodora* (BC), which is also known as “lemon myrtle”, “lemon-scented myrtle”, or “lemon ironwood”, is a species native to Australia. It belongs to a shrub of the Myrtaceae family. Numerous biological and pharmacological actions, such as antioxidant [29], antibacterial [30], and antifungal [31] actions, have been studied using EOs derived from several species of the Myrtaceae family. The fresh or dried leaves of BC have been widely used as ingredients in food flavorings, perfumes, and personal care products in Australia [32]. This Australian native plant has recently received increasing attention in Malaysia, due to its unique flavor, fragrance, and biological properties. Methanolic and ethanolic leaf extracts of BC have been reported, in several studies, to possess good antioxidant, antimicrobial, and anti-inflammatory activities [33,34]. It has been shown that BCEO is a rich source of citral, which possesses versatile biological activities, such as antimicrobial, antifungal, antiviral, and antioxidant activities [35]. 

However, the chemical composition of EOs can vary remarkably with extrinsic conditions, such as geographic origin and climate conditions. Additionally, few studies have been carried out on the antibiofilm activity of BC, and only two studies reported on the antibiofilm activity of aqueous and ethanolic leaf extracts of BC against *Pseudomonas aeruginosa* and *Streptococcus mutans* [36,37]. Therefore, this study aimed to evaluate the chemical composition and antioxidant activity of BCEO from Malaysia, as well as its antibacterial and antibiofilm activities against four human pathogenic bacteria: *Staphylococcus aureus*, *Staphylococcus epidermidis*, *Escherichia coli*, and *Klebsiella pneumoniae*. To the best of our knowledge, this is the first report on the antibiofilm activity of BCEO against four common nosocomial pathogens.

## 2. Results 

### 2.1. Chemical Composition

BCEO was analyzed by GC–MS and presented in Figure 1. The chemical compositions of BCEO are listed in detail in Table 1, along with their retention times (Rt), molecular formulae, compound groups, Kovats indices (KI), and percentages (trace components below 0.1% are not listed). Twenty-one compounds, corresponding to 98.50% of the total oil, were identified and quantified in BCEO, which contains a complex mixture of compounds. GC–FID and GC–MS analysis showed that BCEO obtained from the leaves were separated into four compound classes: oxygenated monoterpenes (91.95%), monoterpene hydrocarbons (0.22%), sesquiterpene hydrocarbons (0.33%), and others (6.00%) (Table 1). The oxygenated monoterpenes were the most abundant phytocompounds in BCEO, dominated by geranial phytocompounds (52.13%) and neral phytocompounds (37.65%). The monoterpene hydrocarbon in BCEO was dominated by β-Myrcene (0.22%), while the predominant sesquiterpene hydrocarbons were Germacrene B (0.18%) and α-Gurjunene (0.15%) (Table 1).

### 2.2. Antioxidant Activity of BCEO

In this study, BCEO exhibited considerable antioxidant activity, based on both the DPPH method and the FRAP method, compared with that of standard antioxidant ascorbic acid. BCEO had an IC_50_ value of 42.57 µg/mL for DPPH radical scavenging activity, while the IC_50_ value of ascorbic acid was 7.23 µg/mL. Using the FRAP method, BCEO had an EC_50_ value of 20.03 µg/mL, which was approximately 1.49 times the value obtained for ascorbic acid (13.48 µg/mL).

### 2.3. Antibacterial Activity of BCEO

#### 2.3.1. Antibacterial Activity Analyzed by the Disk Diffusion Method

In this study, BCEO had an inhibition effect on the four tested bacteria using the agar disk diffusion method, as shown in Table 2. BCEO exhibited significantly (*p* < 0.05) higher antibacterial activity, as indicated by the larger inhibition zone, against Gram-positive bacteria (*S. aureus*, 31.13 ± 0.29 mm; *S. epidermidis*, 50.17 ± 0.29 mm) than that of gentamicin (*S. aureus*, 22.30 ± 0.60 mm; *S. epidermidis*, 24.30 ± 0.60 mm). No significant difference was found between BCEO and gentamicin in the antibacterial activity against *E. coli*. However, the results revealed that the antibacterial effect of BCEO against *K. pneumoniae* was significantly (*p* < 0.05) lower than that of gentamicin. The negative control, 10% DMSO, showed no inhibitory effect against any of the four tested bacteria.

#### 2.3.2. Antibacterial Activity Analyzed by the Broth Dilution Method to Determine Minimum Inhibitory Concentration (MIC) and Minimum Bactericidal Concentration (MBC)

The minimum inhibitory concentration (MIC) and minimum bactericidal concentration (MBC) of BCEO against the four bacterial strains were determined, as shown in Table 3. It was observed that BCEO exhibited bacteriostatic and bactericidal activities against all four bacterial strains. The lowest MIC of BCEO was found against *S. aureus* and *S. epidermidis* (6.25 μL/mL), followed by *E. coli* and *K. pneumoniae* (12.5 μL/mL). The MIC values of BCEO against *S. aureus* and *S. epidermidis* were lower than those of the standard antibiotic streptomycin. However, the MIC values of *E. coli* and *K. pneumoniae* were higher than that of streptomycin. The BCEO concentration of 50 µL/mL was determined as the MBC for all four tested bacteria. Overall, the results indicated that the MBC values of both BCEO and streptomycin against the four pathogenic bacteria strains were higher than their MIC values. Bacterial growth was observed in all assays for both the negative control (10% DMSO) and the growth control, confirming the non-bacterial inhibition of 10% DMSO and the viability of the bacteria strains used, and demonstrating the antibacterial effect of BCEO.

### 2.4. Antibiofilm Activity of BCEO

#### 2.4.1. Biofilm Inhibitory Activity

The biofilm inhibitory activity of BCEO was evaluated on the four bacteria strains at different concentrations, 0.5×, 1× and 2× MICs. Figure 2 summarizes the percentage of biofilm inhibition when different concentrations of BCEO were used. Overall, it was found that BCEO significantly (*p* < 0.05) inhibited 90.01% to 93.39%, 95.79% to 96.44%, and 85.10% to 91.14% of the *S. aureus*, *S. epidermidis,* and *E. coli* biofilms, respectively, at the three tested concentrations, compared with the standard antibiotic streptomycin (*S. aureus*, 74.78% to 85.77%; *S. epidermidis*, 72.00% to 86.40%; *E. coli*, 67.65% to 83.17%). However, the biofilm inhibitory ability of BCEO (86.98% to 88.79%) against *K. pneumoniae* was comparable to that of streptomycin (93.36% to 94.29%) at 0.5×, 1×, and 2× MICs.

#### 2.4.2. Biofilm Eradication Activity

A modified microdilution assay was used to investigate the ability of BCEO in eradicating the established biofilms of *S. aureus*, *S. epidermidis*, *E. coli*, and *K. pneumoniae* at 0.5×, 1×, and 2× MICs. The effect of BCEO on the eradication of established biofilms of tested bacterial strains at 0.5×, 1×, and 2× MICs, expressed as biofilm eradication (%), are illustrated in Figure 3A, Figure 3B and Figure 3C, respectively. Similarly, BCEO exhibited a significantly (*p* < 0.05) stronger ability to eradicate pre-formed biofilms of *S. aureus* (70.92% to 81.28%), *S. epidermidis* (83.56% to 90.73%), and *E. coli* (74.62% to 85.80%) than that of streptomycin (*S. aureus*, 56.95% to 72.50%; *S epidermidis*, 49.97% to 78.62%; *E. coli*, 50.98% to 77.69%) at 0.5×, 1×, and 2× MICs. There was no significant difference observed between BCEO (61.00% to 83.99%) and streptomycin (66.76% to89.73%) in the biofilm eradication activity against *K. pneumoniae* at 0.5×, 1×, and 2× MICs.

## 3. Discussion 

In the present study, the chemical composition of BCEO was rich in citral, a mixture of two geometric isomers: an E-isomer, known as citral A or geranial (52.13%), and a Z-isomer, known as citral B or neral (37.65%), which constituted 89.78% of the total composition. Various studies have identified the chemical composition of BCEO from different countries. The BCEO composition results of this study are approximately consistent with those of de Andrade Santiago et al. [38], who reported that the basic compounds were citral (91.19%), made from a combination of the isomeric aldehydes geranial (39.82%) and neral (51.37%). A previous study carried out in Australia by Southwell et al. [39] also found that BCEO was composed of geranial (46.1% to 60.7%) and neral (32.0% to 40.9%) as the major compounds, followed by iso-geranial (1.0% to 4.2%), iso-neral (0.6% to –2.7%), linalool (0.3% 1.0%), 6-methyl-5-hepten-2-one (0.1% to 2.5%), citronellal (0.1% to 0.9%), and myrcene (0.1% to 0.7%). Similarly, Daimo [40] showed that geranial (44% to 49%) and neral (37% to 39%) were the most important ingredients in BCEO from Australia. A study conducted by Kean et al. [41] in Malaysia revealed that volatile oil acquired from the leaves was comprised of predominantly neral (39.57%) and geranial (52.43%), followed by linalool (0.37%), citronellal (0.15%), E-isocitral (2.47%), geraniol (0.66%), ethyl geranate (0.13%), spathulenol (0.09%), and caryophyllene (0.08%). The quantitative and chemical composition variations of BCEO could be attributed to several factors, such as the anatomical and physiological characteristics of the plant, the geographical position, the ecological conditions, the period of collection, the storage conditions, and the oil extraction technique [42,43,44].

The method for measuring antioxidant activity is still evolving. In this study, the antioxidant activity of BCEO was determined by two different methods: DPPH and FRAP. Although the expression of antioxidant results has not been standardized, the DPPH and FRAP methods can be used to determine the antioxidant capacity of EOs [45,46]. The DPPH method is one of the most simple, acceptable, and widely used antioxidant methods for evaluating the radical scavenging activity of organic compounds. It has typically been used for plant extracts, EOs, and isolated organic substances [47]. Additionally, DPPH is a stable free radical that is used to simulate the antioxidant activity of chemical constituents of EOs, extracts, and other substances from natural products [48]. The purple DPPH free radical transforms into a yellow stable molecule when the odd electron of the DPPH radical is paired with hydrogen from a free radical scavenging antioxidant. Hence, the degree of DPPH discoloration may indicate the scavenging potential of the antioxidant extract. 

The FRAP assay was applied in this study to evaluate the reducing power of EOs. For the FRAP method, the presence of reducing agents in the plant extract leads to the reduction of a colored pale yellow ferric-tripyridyltriazine (Fe^3+^-TPTZ) complex to a blue-colored ferrous tripyridyltriazine (Fe^2+^-TPTZ) [49]. The reducing capacity is related to the degree of hydroxylation and the degree of conjugation of the bonds that are present in the phenolic compounds of EOs and extracts [50]. At low cost, this method was useful for screening the reducing capacities and comparing the efficiencies of different compounds, such as those in EOs [51]. The results of the study showed that BCEO possesses the activity to reduce the free radicals of DPPH and Fe^3+^-TPTZ. The great antioxidant activity of BCEO could be attributed to the effect of its key compounds, geranial and neral (two isomers of citral), as they were the significant oxygenated monoterpenes identified in the current study. Furthermore, minor compounds, such as monoterpene hydrocarbons and sesquiterpene hydrocarbons in BCEO, can enhance antioxidant activity. These chemical compounds of EO might act individually or synergistically as antioxidants. Several studies have indicated that enhanced antioxidant activity was found in EOs which were rich in oxygenated monoterpenes [52,53]. It has been reported that the antioxidant capacity of oxygenated compounds could be ascribed to the free electrons, due to high oxygenation [54,55]. The major compounds, geranial and neral, have been reported in high levels in EO of *Cymbopogon citratus* (lemongrass) (70% to 85%) [56] and *Lippia alba* (70.6% to 79.0%) [57], exhibiting strong antioxidant activity [58,59,60]. Recent studies have associated antioxidant activity to geranial and neral by demonstrating that these compounds are active in scavenging ROS [61,62,63]. Furthermore, citral is known to exhibit various medicinal properties, including inhibiting oxidant activity, cyclooxygenase-2 (COX-2) expression, and nuclear factor kappa B (NF-kB) activation [64]. A study carried out by Bouzenna et al. [65] demonstrated the antioxidant effects of citral in rat small intestine epithelial cells (IEC-6 cells), indicating that citral can protect against aspirin-induced oxidative stress. Therefore, BCEO can be a promising source of natural antioxidant with its high level of oxygenated monoterpenes (citral).

In this study, BCEO displayed considerable antibacterial activity against Gram-positive and Gram-negative bacterial isolates. Four bacterial strains, *S. aureus*, *S. epidermidis*, *E.coli*, and *K. pneumoniae*, were used in this study, as they are common nosocomial pathogens and often form biofilms on medical devices [66]. BCEO showed varying inhibitory activity on the four tested bacterial strains, in the following order: *S. epidermidis* > *S. aureus* > *E. coli* > *K. pneumoniae*. The high antibacterial activity of BCEO against *S. aureus* is strongly correlated with the results of Wilkinson et al. [67], which demonstrated a significant antibacterial effect of two different BCEO samples against *S. aureus* with inhibition zones ranging from 11.50 mm to 32.00 mm. Wilkinson et al. also stated that the two BCEO samples exhibited antibacterial activity against *E. coli* in the range of 8.00 mm to 16.50 mm. These inhibition zones of BCEO against *E. coli* were lower than those of this study. 

Another previous study investigated 91 essential oils and found that BCEO exhibited the largest inhibition zone (65.00 mm) against methicillin-resistant *Staphylococcus aureus* (MRSA), compared with the other EOs that were tested [68]. Recently, da Silva Júnior et al. [69] found that a citral-chemotype from *Lippia alba* EO displayed strong antibacterial activity against *S. epidermidis* with an inhibition zone of >40 mm and MIC at 2.50 µL/mL. A MIC study carried out by Hayes and Markovic [70] in Australia revealed that BCEO and 100% citral displayed compatible activity against *S. aureus*, MRSA, *E. coli*, *K. pneumoniae*, *Pseudomonas aeruginosa*, *Candida albicans*, and *Aspergillus niger* in the range of 0.05% to 2.00% *v/v*. Their study observed that *S. aureus* was the only organism that showed greater susceptibility to citral. They also showed that the MIC results of BCEO in their study were considerably lower than those of tea tree oil against all of the microorganisms that were tested. Similarly, a recent study indicated that BCEO exerted significant antimicrobial activity against two foodborne pathogenic bacteria, *S. aureus* DSM 1104 and *E. coli* DSM 1103, with MIC values of 50 µg/mL and 200 µg/mL, respectively [71]. Recently, Beikzadeh et al. [72] revealed that BCEO exhibited high antimicrobial activity in inhibiting the growth of pathogenic and spoilage bacteria at MICs ≤ 0.125 and 0.06 μL/mL for *E. coli* and *S. aureus*, respectively. However, their findings were lower than the MICs found in this study, where BCEO showed strong antimicrobial activity at MICs ≤ 12.5 and 6.25 μL/mL for *E. coli* and *S. aureus*, respectively. These variations may be attributed to several factors, such as extraction methods, time, temperature, plant parts, and agroclimatic conditions. Furthermore, although the types of expression of MIC values are variable in the studies, all authors identified great inhibitory activity of BCEO, which corresponds with our findings.

Previous studies showed that the antibacterial activities of EOs from different plant species were mainly associated with their chemical compositions, especially oxygenated monoterpenes, which usually occur in high amounts [73,74,75]. In the present study, the results obtained for antibacterial activity correspond with the great antioxidant activity of BCEO. Based on this observation, it can be speculated that the strong antibacterial activity exerted by BCEO can be attributed to the high amounts of oxygenated monoterpenes, especially geranial and neral. Citral, or one of its isomers (geranial or neral), has been reported to be a molecule that inhibits the growth of several pathogenic and food-spoilage bacteria, such as *S. aureus*, *E. coli, Salmonella enterica*, and *Bacillus cereus* [76,77]. Chueca et al. [78] demonstrated that oxygenated monoterpenes of citral (a mixture of geranial and neral) exhibited antimicrobial activity by causing oxidative stress in *E. coli*. Similarly, a recent study highlighting the antimicrobial activity of BCEO corresponded directly to the high citral content, which was an isomeric mixture of neral (Z-isomer) and geranial (E-isomer) [79]. Meanwhile, the contribution of other minor chemical components in BCEO in intensifying antimicrobial activity, such as β-myrcene and linalool, in combination with citral, has also been reported [80,81,82]. Several studies have revealed that the antimicrobial activity of oxygenated monoterpenes acts by disrupting the microbial cytoplasmic wall, which improves cell permeability and allows the passage of large protons and ions, leading to cell death [83,84]. Shi et al. [85] confirmed that citral exhibited antimicrobial activity against *Cronobacter sakazakii*, an opportunistic Gram-negative bacterium, by causing changes in ATP concentration, cell membrane hyperpolarisation, and a reduction in the cytoplasmic pH. Recently, Zhang et al. [84] reported that citral could change a cell’s shape by destroying the cell wall and the membrane structure, leading to the loss of cytoplasm and the intracellular leakage of protein, nuclei acid, and other cell substances, as well as influencing the pH balance inside and outside the cell. Therefore, the strong antibacterial activity of BCEO, as indicated in this study, could be due to the high amounts of oxygenated monoterpenes of citral, which contains a mixture of geranial and neral.

Microorganisms can attach to surfaces and produce extracellular polysaccharides, resulting in biofilm formation as a microbial survival strategy, especially in adverse conditions. The National Institutes of Health (NIH) estimated that biofilm formation is responsible for 65% of microbial diseases and more than 80% of chronic infections [86]. Biofilms have great significance for public health because of the increased resistance of biofilm-associated microorganisms to antimicrobial agents. Biofilm formation is strongly associated with indwelling or implanted medical devices, such as urinary catheters, endotracheal tubes, enteral feeding tubes, ventilators, orthopaedic implants, and prosthetic joints, causing medical treatments to become increasingly difficult. Therefore, the use of a new natural compound to inhibit or eradicate biofilms is of great importance. In the present study, BCEO has been shown to exhibit dual actions in preventing and eradicating biofilm formation in all of the tested bacterial strains. The biofilm inhibition and eradication abilities of BCEO on the four tested bacterial strains in this study are classified in the following decreasing order: *S. epidermidis* > *S. aureus* > *E. coli* > *K. pneumoniae*. This observation corresponds with this study’s results on antibacterial activity. It was found that BCEO demonstrated significant biofilm inhibition activity (85.10% to 96.44%) and eradication activity (70.92% to 96.44%) against *S. aureus*, *S. epidermidis*, and *E. coli*, compared with those of the standard antibiotic (streptomycin), at 0.5× to 2× MICs. These results agree with those of Duarte et al. [87], who revealed that coriander EO inhibited at least 85% of biofilm formation and eradicated up to 97.14% of 48 h pre-formed biofilms of *Acinetobacter baumannii*. However, the biofilms of *K. pneumoniae* appeared to be less sensitive to being inhibited or eradicated by BCEO than by streptomycin in this study. Several recent reports revealed that the bacteria biofilm could be inhibited or eradicated effectively by EOs, including clove basil oil [88], tea tree oil [89], cedar oil [90], and garlic and thyme oil [91]. Recently, Cáceres et al. [92] reported that EO from *Lippia origanoides* showed detrimental effects against biofilm formation in *E. coli* O33, *E. coli* O157:H7, and *S. epidermidis* ATCC12228, with percentages of biofilm inhibition of 75%, 73%, and 74%, respectively. These biofilm inhibition results were generally lower than our results, which may be due to different plant chemotypes, as *L. origanoides* EO is rich in thymol-carvacrol, while BCEO is rich in citral. Another recent study found that *Origanum majorana*, *Rosmarinus officinalis* and *Thymus zygis* EOs demonstrated high biofilm inhibition and eradication activities against MRSA clinical isolates, with the percentages of inhibition in the range of 10.20% to 95.91%, and the percentages of eradication ranging from 12.65% to 98.01% [93].

The antibiofilm activity of EOs have been investigated in the past two decades; however, there is limited information about the antibiofilm effects of BCEO on pathogenic bacteria. The biofilm inhibition effects of BCEO found in the present study suggest that the addition of EOs prior to biofilm formation eliminates planktonic cells, rendering the abiotic (polystyrene) surface less susceptible to cell adhesion. Furthermore, the biofilm inhibitory effects of BCEO identified in this study can be explained by the modification of bacterial surface proteins, due to the effect of their interactions with EOs in reducing the adhesion of planktonic cells to surfaces, which is the initial attachment phase [94,95]. Several studies have demonstrated that EOs could penetrate the exopolysaccharide matrix of established biofilms and destabilize them, due to their strong intrinsic antimicrobial actions [95,96]. The significant biofilm inhibition and eradication activities of BCEO can also be explained by the fact that the major constituent in this oil, citral, has an effect on the biofilm formation process. 

The biofilm inhibition ability of citral and citral-rich EO from three *Lippia alba* (LAEO) specimens was demonstrated by Porfírio et al. [97], who found a 100% inhibition of *S. aureus* biofilm formation at the concentration of 0.5 mg/mL of citral and all three LAEOs (LA1EO-LA3EO). They also reported that the biofilm elimination ability of citral, LA1EO, LA2EO, and LA3EO were confirmed at concentrations of 0.5 mg/mL, 1 mg/mL, 2 mg/mL, and 2 mg/mL, respectively. A recent study reported that citral applied individually or in combination with linalool, eugenol, and thymol, could exert growth inhibition in planktonic cells and a bacteriostatic effect on the biofilm cells of *Shigella flexneri* [98]. A study on the biofilm inhibition activity of an ethanolic extract of *B. citriodora* by Almousawi et al. [37] showed that this plant extract was effective in inhibiting ten clinical isolates of *P. aeruginosa* that were isolated from various burn sites. Gao et al. [96] indicated that EO of lemongrass (*Cymbopogon flexuosus*) (LGEO) and its major component citral exhibited strong antibiofilm action against mono- and dual-species biofilms formed by *S. aureus* and *Candida species* at low concentrations ranging from 0.0156% to 0.1563%. Gao et al. stated that the biofilm biomass and the cell viability of *S. aureus* and *Candida* spp. (*C. albicans* and *C. tropicalis*) were reduced after exposure to LGEO and citral in a biofilm staining and viability test. The microscopic examinations of these pathogenic bacteria and yeast found that LGEO and citral interfered with the adhesive characteristics of each species and disrupted the exopolysaccharide of biofilm matrix by counteracting carbohydrates, proteins, and nuclei acids in the biofilm. Additionally, the transcriptional analyses revealed that citral reduced the expression of genes involved in quorum sensing, fatty acids, and peptidoglycan biosynthesis in *S. aureus*, and downregulated the virulence factors and hyphal of *C. albicans*. In another study, *C. citratus* EO, citral, and geraniol were found to inhibit the planktonic growth of *E. coli* O157:H7 (MIC = 2.2 mg/mL, 1.0 mg/mL, and 3.0 mg/mL, respectively), the bacterial adhesion (2.0, 2.0, and 4.0 mg/mL, respectively), and the glucan production on stainless steel surfaces. They suggested that this biofilm inhibition effect could be associated with an uncompetitive inhibition of glucosyltransferase activity that was caused by citral and geraniol [99].

The present study also found that Gram-negative bacteria are more resistant to BCEO than Gram-positive bacteria. These results were correlated with a study conducted by Man et al. [100], who showed that the six studied EOs were more susceptible to Gram-positive cocci, such as MRSA, while being more resistant to Gram-negative bacilli including *Pseudomonas aeruginosa*. Several recent studies also indicated that Gram-positive bacteria are more sensitive than Gram-negative bacteria to EOs [83,101]. It has been documented that different compositions of the cell wall in Gram-positive and Gram-negative bacteria may be the key contributors to various inhibition zones, MICs, MBCs, and antibiofilm activities. In fact, the cell wall structure of Gram-negative bacteria is more complex than that of Gram-positive bacteria [102,103]. The outer membrane of Gram-negative bacteria containing hydrophilic lipopolysaccharides could create an effective barrier for hydrophobic compounds, such as those found in EOs [104]. The structure of the Gram-positive bacteria cell wall enables hydrophobic molecules to penetrate the cells and act on both the cell wall and the cytoplasm. Thus, the hydrophobic phenolic compounds present in EOs generally exhibit antimicrobial activity against Gram-positive bacteria [102]. It can be postulated that the strong antimicrobial and antibiofilm effect of BCEO against Gram-positive bacteria in this study may be due to the high amount of hydrophobic citral.

## 4. Materials and Methods

### 4.1. Plant Material Collection and Identification

*Backhousia citriodora* (BC) leaves were obtained from an organic lemon myrtle plantation area in Kuala Linggi, Malacca, Malaysia. The taxonomic identification of BC was confirmed by the botanist from the Biodiversity Unit, Institute of Bioscience (IBS), Universiti Putra Malaysia (UPM), Malaysia. The voucher specimen (MFI 0145/19) was deposited at the Biodiversity Unit, IBS, UPM.

### 4.2. B. citriodora Essential Oil (BCEO) Extraction

Fresh leaves of BC were air-dried for three days and grounded into powder in a grinding machine. BCEO was extracted by hydrodistillation in a Clavenger-type apparatus for 4 h (2 kg of BC leaves in 5 L of distilled water) to obtain the EO. The moisture in the EO was removed by using anhydrous magnesium sulphate (MgSO_4_). The EO obtained was then transferred to sealed dark vials and stored at 4 °C for subsequent analysis.

### 4.3. Gas Chromatography–Flame Ionization Detection (GC–FID) and Gas Chromatography–Mass Spectrometry (GC–MS) Analyses of the BCEO

The GC–MS for the analyses of the volatile constituents of BCEO was performed on an Agilent 7890A/5975C series gas chromatography–mass spectrometry (GC–MS) system (Agilent Technologies Inc., Santa Clara, CA, USA). The system was equipped with an HP-5MS fused silica capillary column (30 m long × 0.25 mm i.d. × 0.25 μm film thickness, Agilent Technologies, Santa Clara, CA, USA). The ultra-pure helium was used as a carrier gas at the flow rate of 1 mL/min with a spilt ratio of 50:1. The oven temperature was programmed from 60 °C for 10 min to 230 °C at a rate of 3 °C/min for 1 min. The injector port temperature was 250 °C. The mass spectra of the BCEO compounds were acquired by electron ionisation (EI) at 70 eV. Data acquisition was performed in full scan mode, using a spectral range of *m*/*z* 50–550. The temperatures of the auxiliary heating zone, the ion source, and the MS quadrupole were 280 °C, 230 °C, and 150 °C, respectively. The GC–FID was carried out for the quantitative analyses on Shimadzu GC–2010 gas chromatography (Shimadzu, Tokyo, Japan), using a fused silica capillary column Zebron ZB-5MS (30 m long × 0.25 mm i.d. × 0.25 µm film thickness) (Phenomenex, Inc., Torrance, CA, USA). The sample injection volume was 0.5 µL. The oven temperature was programmed at 60 °C for 10 min to 230 °C at a rate of 3 °C/min for 1 min. The detector temperature was 250 °C. The carrier gas was ultra-pure helium at a flow rate of 1.9 mL/min, equipped with FID with a spilt ratio of 50:1. The chemical compounds in BCEO were identified by retention indices, calculated using linear interpolation relative to retention times of C_8_–C_23_ of *n*-alkanes standards, and compared to reference spectra (Adams and NIST databases). The percentages composition of BCEO were calculated using the peak area normalization method without correction factors. The measurement was conducted in triplicate.

### 4.4. Measurement of Antioxidant Activity of BCEO

#### 4.4.1. DPPH (2,2-diphenyl-1-picrylhydrazyl) Radical Scavenging Assay

The DPPH free radical scavenging activity of BCEO was determined by following the method described by Semiz et al. [105]. The stock solution (1 mg/mL) of BCEO was prepared in methanol. Dilutions were made to obtain concentrations of 1000, 500, 250, 125, 62.5, 31.25, 15.62, 7.81, 3.90, 1.99, and 0.97 µg/mL. One mL of the diluted solution was mixed with 1 mL of 0.2 mM of DPPH (2,2-diphenyl-1-picrylhydrazyl) methanolic solution. After 30 min of incubation in the dark at room temperature, the absorbance was recorded at 517 nm against a blank (methanol solution). The control samples contained all the reagents except BCEO. All measurement was performed in triplicate. Ascorbic acid was used as the standard. The DPPH radicals scavenging activity was calculated using the following equation:%inhibition = [(A_control_ − A_sample_)/A_control_] × 100
where A_control_ = the absorbance value of the control and A_sample_ = the absorbance values of the DPPH radical in the presence of BCEO. The inhibition percentage was plotted against the sample concentrations, and 50% of the inhibitory concentration (IC_50_) of the DPPH values were defined by linear regression analysis.

#### 4.4.2. Ferric Reducing Antioxidant Power Assay (FRAP)

The total antioxidant activity was measured by the ferric reducing antioxidant power assay (FRAP), following the method described by Vijayalakshmi et al. [106]. In brief, different concentrations of BCEO (0.97–1000 µg/mL) were added to 2.5 mL of 0.2 M of sodium phosphate buffer (pH 6.6) and 2.5 mL of 1% potassium ferricyanide [K_3_Fe(CN)_6_] solution. Then, the reaction mixture was vortexed and incubated at 50 °C for 20 min using a vortex shaker. After incubation, 2.5 mL of 10% trichloroacetic acid was added to the mixture and centrifuged at 3000 rpm for 10 min. After that, 2.5 mL of the supernatant was mixed with 2.5 mL of distilled water and 0.5 mL of 0.1% ferric chloride. Ascorbic acid was used as a reference standard. The colored solution was read at 700 nm against blank with reference to standard using a UV-Spectrophotometer. Blank for each solvent was run using the same procedure but replacing BCEO with an equal volume of solvent. The experiment was conducted in triplicate. The EC_50_ value (µg/mL) was BCEO concentration at the absorbance 0.5 (a.u.) for the reducing power and was calculated from the graph of absorbance at 700 nm against extracts and EO concentrations.

### 4.5. Determination of Antibacterial Activity of BCEO

#### 4.5.1. Microorganisms and Medium

The antibacterial activity of BCEO was evaluated against four clinical bacteria strains (*Staphylococcus aureus*, *Staphylococcus epidermidis*, *Escherichia coli*, and *Klebsiella pneumoniae*) in this study. These clinical strains were obtained from the Universiti Kebangsaan Malaysia Medical Centre (UKMMC), Wilayah Kuala Lumpur, Malaysia.

#### 4.5.2. Agar Disk Diffusion Method

The antibacterial activity of BCEO was measured using an agar disk diffusion assay, as described in a previous work [107], with minor modifications. Before antimicrobial testing, each bacteria strain was subcultured twice onto nutrient agar (NA) plates and incubated for 24 h at 37 °C to obtain colonies. After overnight incubation, colonies of each bacteria strain were taken from their respective cultures with a sterile inoculating loop and transferred to a sterile physiological saline glass tube and vortexed thoroughly. The turbidity of each bacterial suspension was then compared with that of the 0.5 McFarland standard solution (containing about 1.5 × 10^8^ CFU/mL). BCEO was dissolved in 10% dimethyl-sulfoxide (DMSO) to 10 mg/mL. Each adjusted bacterial suspension was inoculated on Muller–Hinton agar (MHA) plates and allowed to dry for 5 min. The blank sterile antibiotic disks (diameter = 6 mm) were impregnated with 30 µL of BCEO. The BCEO-containing disks were then aseptically placed on the inoculated MHA plates. Gentamicin (30 µg) was used as the positive control, while a 10% DMSO-soaked disk was used as the negative control. The plates were incubated for 24 h at 37 °C. The diameters of the inhibition zones were measured and recorded in mm after incubation. All assays were conducted in triplicate in three independent experiments.

#### 4.5.3. Determination of the Minimum Inhibitory Concentration (MIC) and the Minimum Bactericidal Concentration (MBC)

The antibacterial activities of BCEO were investigated using the microdilution method in 96-well plates to determine MIC values, as described in an earlier work [107], with minor modifications. This test was carried out with *S. aureus*, *S. epidermidis*, *E. coli*, and *K. pneumoniae*. Each bacterium was prepared and adjusted to 0.5 McFarland (containing about 1.5 × 10^8^ CFU/mL). BCEO was dissolved in 10% DMSO to adequate concentrations and then syringe-filtered through 0.22 µm nylon membrane filters. Two-fold serial dilutions were conducted to yield volumes of 100 μL per well, with final concentrations ranging from 0.024 µL/mL to 50 µL/mL in the MHB medium. An aliquot of 50 µL of the bacterial dilution was added to each well and the final volume of 200 µL/well was adjusted with the MHB medium. As a negative control, an aqueous solution of 10% DMSO was used. Then, 50 μL of the inoculum solution was added to each of the wells. The wells containing only MHB medium with inoculum and the wells containing MHB medium, inoculum, and streptomycin were used as growth and positive controls, respectively. Microplates were incubated for 24 h at 37 °C. After overnight incubation, bacterial growth was evaluated with resazurin. Resazurin at a concentration of 0.015% was prepared, using sterile distilled water, and filtered. An aliquot of 30 µL of 0.015% resazurin was added to each well. The plates were incubated for 30 min in the dark. The viable bacteria were detected by the change of blue-purple to pink. The MIC was defined as the lowest concentration that completely inhibited any visual growth of the tested bacteria. To determine the MBC values, an aliquot of 10 µL from each well that did not show an apparent growth as confirmed by MIC determination were plated and streaked on an NA agar plate. The plates were incubated at 37 °C for 24 h. The MBC was determined as the lowest concentration where no growth was visually observed. All assays were conducted in triplicate in three independent experiments.

### 4.6. Assessment of Antibiofilm Activity of BCEO

#### 4.6.1. Preparation of Bacterial Cultures

The four bacterial cultures (*S. aureus*, *S. epidermidis*, *E. coli*, and *K. pneumoniae*) were cultured by streaking onto sterile tryptone soy agar (TSA) and incubated at 37 °C for 24 h. After incubation, the bacteria were inoculated in sterile tryptone soy broth (TSB) and incubated in a shaking incubator at 37 °C for 24 h. The overnight bacterial culture was standardized to a concentration of 1.0 × 10^6^ CFU/mL prior antibiofilm assays. This was achieved by diluting the overnight culture with TSB supplemented with 2% glucose to obtain an absorbance of 0.02 at OD_590_ nm, using a spectrophotometer [102].

#### 4.6.2. Inhibition of Biofilm Formation, Prevention of Initial Bacteria Cell Attachment

BCEO was evaluated for its potential anti-adhesion properties on a 96-well microplate, as described by Sandasi et al. [108] with minor modifications. One hundred µL of standardised culture dilution (1.0 × 10^6^ CFU/mL) was dispensed into each well. A volume of 100 µL of BCEO dissolved in TSB, containing 0.5% (*v/v*) of Tween 80, was added to each 96-well microplate to reach final concentrations of 0.5×, 1× and 2× MICs. The wells containing TSB supplemented with 0.5% (*v/v*) of Tween 80 and inoculum without BCEO were used as growth controls. Streptomycin (0.5×, 1× and 2× MIC values) was included as a positive antibiotic control. The microplate was incubated at 37 °C for 24 h without shaking to allow cell attachment and biofilm development. Each experiment was carried out in triplicate in three independent experiments. Following overnight incubation, the modified crystal violet assay was conducted to assess biofilm biomass [106].

#### 4.6.3. Inhibition of Development of Pre-Formed Biofilms—Evaluation of Eradication of Biofilm Mass

The effects of BCEO on established biofilm were evaluated using the procedure as described by Jardak et al. [44], with slight modifications. Briefly, biofilm formation was achieved by aliquoting 100 µL of a standardized bacterial culture (1.0 × 10^6^ CFU/mL) into a 96-well microplate and incubated at 37 °C for 4 h. After 4 h incubation, an aliquot of 100 µL of BCEO diluted in TSB supplemented with 0.5 % (*v/v*) Tween 80 was dispensed into each well to yield final concentrations of 0.5×, 1×, and 2× MICs. The microplates were then incubated at 37 °C for 24 h. Streptomycin (0.5×, 1× and 2× MIC values) was included as a positive control. The wells containing TSB supplemented with 0.5% (*v/v*) of Tween 80 and inoculum without BCEO served as growth controls. The biofilm biomass was quantified using the modified crystal violet staining method after incubation. Each experiment was performed in triplicate in three independent experiments.

#### 4.6.4. Modified Crystal Violet Assay

Cell attachment and eradication were assessed using the modified crystal violet (CV) assay, as described by Djordjevic et al. [109]. Following incubation, the microplates were washed three times with sterile distilled water to remove the loosely attached cells. The plates were air-dried, then oven-dried at 60 °C for 10 min. After drying, 200 µL of methanol was added for 15 min, then air-dried for 10 min. Then, 100 µL of 1% CV was added to each well and incubated at room temperature for 15 min. Next, the microplates were washed three times with sterile distilled water to remove excess CV. Finally, 125 µL of ethanol was added to destain the wells and 100 µL of the destaining solution was transferred to a new microtiter plate. The optical density (OD) of each well was measured at 595 nm using a microplate reader. The percentages of adherence inhibition and biofilm eradication were calculated using the following formula:[(OD (growth control) − OD (sample))/OD (growth control)] × 100
where OD growth control refers to the absorbance of the bacteria growth without BCEO and OD sample refers to the absorbance of BCEO or streptomycin with bacteria.

### 4.7. Statistical Analysis

The Statistical Package for the Social Sciences (SPSS) Version 25 was used to carry out the independent samples *t*-test to determine the significant differences between the means. All differences were considered significant at *p* < 0.05.

## 5. Conclusions

This study assessed the chemical composition and the antioxidant, antibacterial, and antibiofilm activities of BCEO. The GC–MS analysis of BCEO revealed the presence of 21 compounds. Oxygenated monoterpenes compounds were predominant, especially geranial and neral (isomers of citral), which constituted 89.78% of the total oil. BCEO exhibited promising antioxidant, antibacterial, and antibiofilm activities. The antibacterial and antibiofilm activities of BCEO against the tested bacteria stains were classified in the following decreasing order: *S. epidermidis* > *S. aureus* > *E. coli* > *K. pneumoniae*. The findings of this study indicated that BCEO was more effective against Gram-positive bacteria than against Gram-negative bacteria. The dual action of BCEO in preventing and eradicating biofilm formation has made this plant a potential candidate for nosocomial infection therapy. Taken together, these findings indicate the potential of BCEO as a promising agent against human pathogenic bacteria and their biofilms. Therefore, BCEO can be a potential candidate for developing a new therapeutic system or a drug adjuvant in the future. Additional research works should be conducted to evaluate the safety and effectiveness of BCEO as an alternative to synthetic drugs.

## Figures and Tables

**Figure 1 molecules-27-04895-f001:**
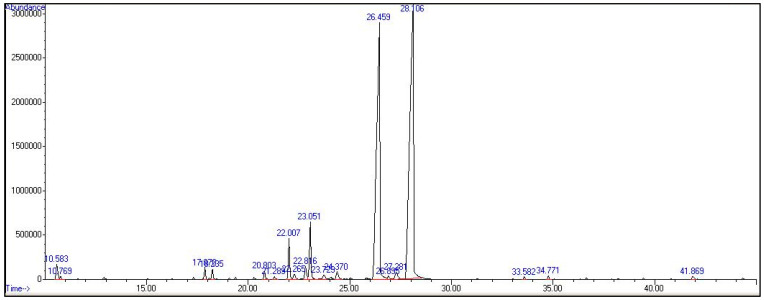
Chromatogram of *B. citriodora* EO (BCEO) compounds derived from GC–MS.

**Figure 2 molecules-27-04895-f002:**
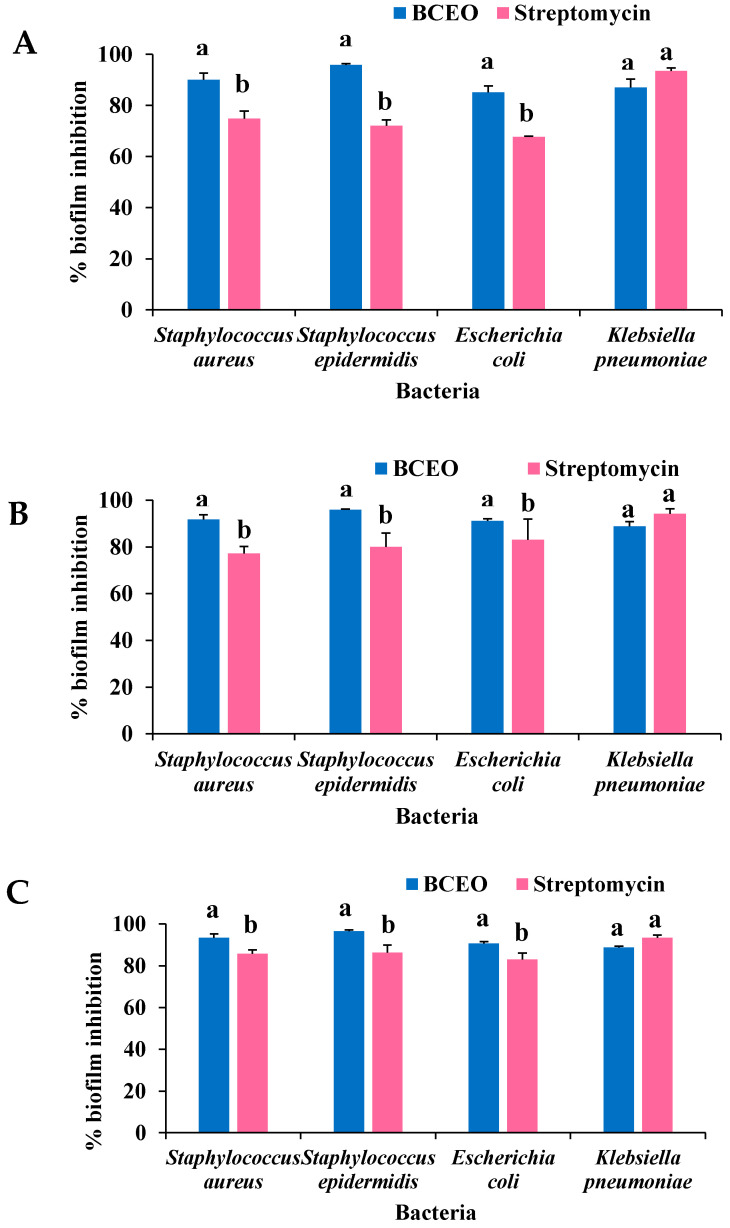
Effects of BCEO and streptomycin on the inhibition of biofilm formation of *S. aureus*, *S. epidermidis*, *E. coli,* and *K. pneumoniae*, expressed as biofilm inhibition (%) at (**A**) 0.5× MIC, (**B**) 1.0× MIC, and (**C**) 2.0× MIC. Bars represent mean ± SD of triplicates in three independent experiments. Error bars are standard deviations. Within the same bacteria, bars with different letters (a, b) differ significantly (*p* < 0.05). BCEO, *B. citriodora* essential oil.

**Figure 3 molecules-27-04895-f003:**
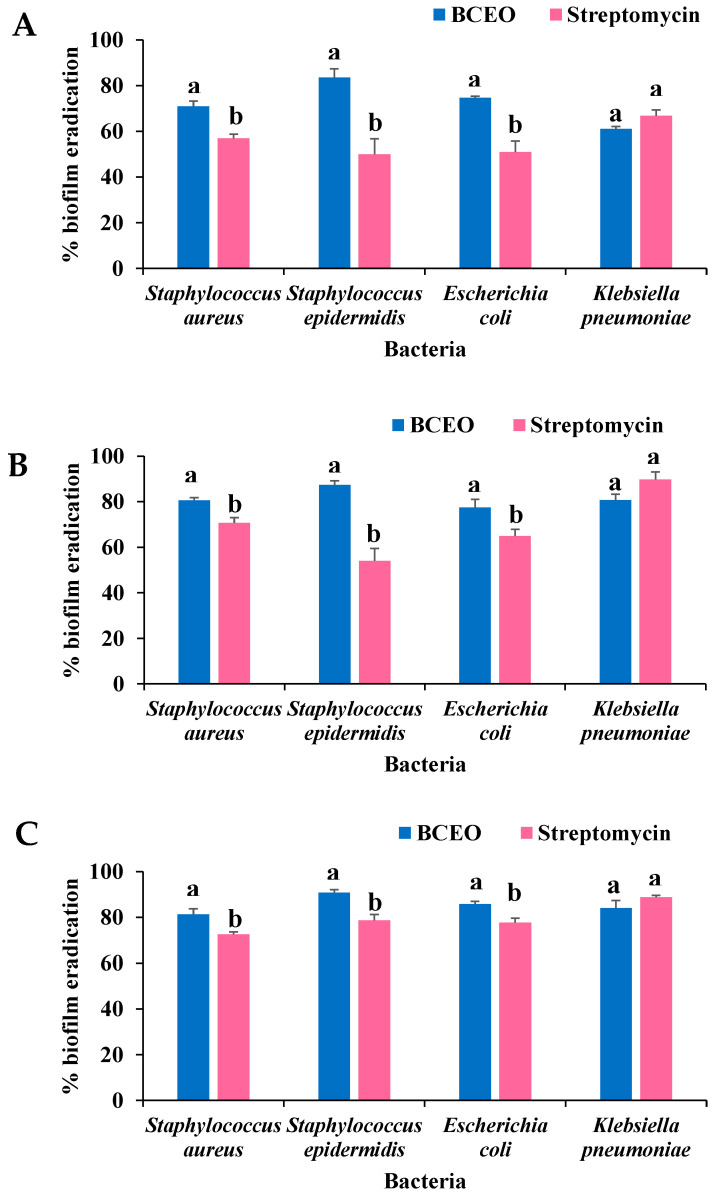
Effects of BCEO and streptomycin on the eradication of pre-formed biofilms of *S. aureus*, *S. epidermidis*, *E. coli*, and *K. pneumoniae*, expressed as biofilm eradication (%) at (**A**) 0.5× MIC, (**B**) 1.0× MIC, and (**C**) 2.0× MIC. Bars represent mean ± SD of triplicates in three independent experiments. Error bars are standard deviations. Within the same bacteria, bars with different letters (a, b) differ significantly (*p* < 0.05). BCEO, *B. citriodora* essential oil.

**Table 1 molecules-27-04895-t001:** Chemical composition of BCEO.

N	Compound ^a^	Molecular Formula	Compound ^b^ Group	RT ^c^	KI ^d^	Area Percentage (%) *
1	6-methyl-5-hepten-2-one	C_8_H_14_O	Other	10.583	994	1.02
2	β-Myrcene	C_10_H_16_	MH	10.769	998	0.22
3	(2-methylprop-1-enyl)-cyclohexa-1,5-diene	C_10_H_14_	Other	12.904	1030	0.12
4	Rosefuran	C_10_H_14_O	OM	17.870	1105	0.39
5	Linalool	C_10_H_18_O	OM	18.235	1110	0.53
6	p-mentha-E-2,8(9)-dien-1-ol	C_10_H_16_O	OM	20.279	1145	0.10
7	Trifluoroacetyl-lavandulol	C_12_H_17_F_3_O_2_	Other	20.803	1154	0.33
8	Citronella	C_10_H_18_O	OM	21.289	1163	0.11
9	Cyclopropene	C_3_H_4_	Other	22.007	1175	1.40
10	α-Phellandren-8-ol	C_10_H_16_O	OM	22.265	1179	0.32
11	Cyclohexane, ethenyl-	C_8_H_12_	Other	23.051	1192	2.72
12	Trans-p-menth-2-en-1,8-diol	C_10_H_18_O_2_	OM	23.729	1204	0.20
13	β-Methylcrotonaldehyde	C_5_H_8_O	Other	24.060	1211	0.18
14	Cis-Carveol	C_10_H_16_O	OM	24.370	1216	0.10
15	Tetracyclo [3.3.0(2,6).0(3,9)]decan-2-ol	C_10_H_14_O	OM	25.032	1229	0.42
16	Neral	C_10_H_16_O	OM	26.459	1256	37.65
17	Geranial	C_10_H_16_O	OM	28.106	1272	52.13
18	Cinnamic acid	C_9_H_8_O_2_	Other	32.551	1379	0.12
19	1-Propanesulfonothioic acid	C_6_H_14_O_2_S_2_	Other	33.582	1383	0.11
20	α-Gurjunene	C_15_H_24_	SH	34.771	1407	0.15
21	Germacrene B	C_15_H_24_	SH	41.869	1556	0.18
	Total					98.50
	OM					91.95
	MH					0.22
	SH					0.33
	Others					6.00

^a^ Compounds listed in order of their elution from an HP-5MS fused silica capillary column; ^b^ MH, monoterpene hydrocarbon; OM, oxygenated monoterpene; SH, sesquiterpene hydrocarbon; ^c^ retention time (minutes); ^d^ Kovats index calculated against relative to C_8_-C_23_
*n*-alkanes for HP-5MS. * Only the two first decimal places are presented.

**Table 2 molecules-27-04895-t002:** Antibacterial activity of BCEO and standard antibiotic (gentamicin) expressed by the diameter of the inhibition zone (mm).

Bacteria	Zone of Inhibition (mm)	
	BCEO	Gentamicin
*Staphylococcus aureus*	31.13 ± 0.29 ^a^	22.30 ± 0.60 ^b^
*Staphylococcus epidermidis*	50.17 ± 0.29 ^a^	24.30 ± 0.60 ^b^
*Escherichia coli*	20.33 ± 0.58 ^a^	20.00 ± 0.00 ^a^
*Klebsiella pneumoniae*	12.67 ± 0.58 ^b^	18.00 ± 0.00 ^a^

Data represent mean ± SD of triplicates in three different independent experiments. ^a,b^ Means within a row with different superscripts differ significantly (*p* < 0.05). BCEO, *B. citriodora* essential oil. Positive control: 30 μg/6 mm disc gentamicin. Negative control: 10% DMSO. All negative controls showed no inhibition.

**Table 3 molecules-27-04895-t003:** Minimum inhibitory concentration (MIC) and minimum bactericidal concentration (MBC) of BCEO and standard antibiotic (streptomycin) against four pathogenic bacteria strains.

Bacteria	BCEO		Streptomycin	
	MIC	MBC	MIC	MBC
*Staphylococcus aureus*	6.25	50.00	15.63	125.00
*Staphylococcus epidermidis*	6.25	50.00	10.00	64.00
*Escherichia coli*	12.50	50.00	7.81	15.63
*Klebsiella pneumoniae*	12.50	50.00	3.91	31.25

Data represent mean of triplicates in three different independent experiments. MIC and MBC are expressed in μL/mL.

## Data Availability

Data are contained within the article.

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
