# Peer review of "Chemical Composition, Antioxidant, Antibacterial, and Antibiofilm Activities of Backhousia citriodora Essential Oil"

_molecules, 2022, doi:10.3390/molecules27154895_

Round 1

Reviewer 1 Report

The work entitled „Chemical Composition, Antioxidant, Antibacterial and Antibiofilm Activities of Backhousia citriodora Essential Oil

The main problems in this paper are as follows:

What are the innovative points of this paper, which should be clearly presented in the abstract and introduction.

Indicate why you choose the selected assays for antioxidant activity;

The same thing do for selected bacterial strains;

The clarity of the pictures in this article needs to be improved; maybe to be colorful for easier observation.

The references cited in this paper are old, it is suggested to quote more references in the past five years.

The format of this article still has some problems and needs to be modified.

Overall the author of this paper has done a good job and recommends that it is accepted after minor revision.

Author Response

Please see the attachment for the responses to Reviewer 1 and the revised manuscript.

Reviewer 2 Report

Dear Authors

  The MS entitled “Chemical Composition, Antioxidant, Antibacterial and Antibiofilm Activities of Backhousia citriodora Essential Oil” were reviewed very critically. Th article is very well written and consistent in reading. The data seems to be original and has been justified with suitable literature. My suggestions are provided below.

Page 1 abstract line 21: essential oils from leaves? Write the part used in obtaining EOs.

Page 3: figure 1: provide clear GC chromatogram. The original may be submitted separately as supplementary file. The Ri/Rt values could be placed on top of each peak.

Page 4: line 145. Correct the ligand spellings (kovats).

Page 12: line 452-458.essential oil extraction: The authors stated that the leaves were dried and powdered. The air drying could affect the whole essential oil components since these are volatile compounds. Did they carry some separate experiments on fresh vs dried leaves?

Page 13: line 542. In triplicate or three times? Corrections needed.

Also check the following literature for your discussion.

Khan, F. A., Jan, A. K., Khan, N. M., Khan, N. A., & Khan, S. (2015). GC/MS analysis, antimicrobial and in vitro anti-cholinesterase activities of the essential oil from Buddleja asiatica. Bangladesh Journal of Pharmacology10(4), 891-895.

Author Response

Please see the attachment for the responses to Reviewer 2 and the revised manuscript.

Reviewer 3 Report

In this work entitled “Chemical Composition, Antioxidant, Antibacterial and Antibiofilm 
Activities of Backhousia citriodora Essential Oil
”, authors assessed the chemical composition, antioxidant, antibacterial and antibiofilm activities of the Backhousia citriodoraEssential Oil. The GC-MS analysis of BCEO revealed the presence of 21 compounds with promising antioxidant, antibacterial as well as antibiofilm activity making this plant a potential candidate for nosocomial infection treatment. 

The manuscript is well written, easy to understand and clearly explained. In my opinion, it can be accepted after minor revision addressing following comments:

·      Figure 1 enlarge the axis of retention time to better distinguish peaks;

·      Table 1 It is not so clear that the percentage of “Identified compounds” is the sum of the “Oxygenated monoterpenes-Monoterpene hydrocarbons-Sesquiterpene; hydrocarbons-Others” percentages. It should be better specified in the table;

·      Results relative to mock control (DMSO) should be mentioned along section results;

·      Table 4 add statistical analysis.

Author Response

Please see the attachment for the responses to Reviewer 3 and the revised manuscript.
